

# Mitochondrial phylogeny and comparative mitogenomics of closely related pine moth pests (Lepidoptera: *Dendrolimus*)

Jie Qin[1],[*], Jing Li[1],[*], Qiang Gao[2], John-James Wilson[3] and Ai-bing Zhang[1]

[1] College of Life Sciences, Capital Normal University, Beijing, P. R. China
[2] Institute of Genetics and Developmental Biology, Chinese Academy of Sciences, Beijing, P. R. China
[3] International College Beijing, China Agricultural University, Beijing, P. R. China
[*] These authors contributed equally to this work.

## ABSTRACT

**Background:** Pine moths, *Dendrolimus* spp. (Lasiocampidae), are serious economic pests of conifer forests. Six closely related species (*Dendrolimus punctatus*, *D. tabulaeformis*, *D. spectabilis*, *D. superans*, *D. houi*, and *D. kikuchii*) occur in China and cause serious damage to coniferophyte. The complete mito genomes of *Dendrolimus* genus are significant to resolve the phylogenetic relationship and provide theoretical support in pest control.
**Methods:** The complete mitogenomes of three species (*D. superans*, *D. houi*, and *D. kikuchii*) were sequenced based on PCR-amplified with universal primers, which were used to amplify initial fragments. Phylogenetic analyses were carried out with 78 complete mitogenomes of lepidopteran species from 10 superfamilies.
**Results:** The complete mitochondrial genomes of these three species were 15,417, 15,381, and 15,377 bp in length, separately. The phylogenetic analyses produced consistent results for six *Dendrolimus* species based on complete mitogenomes, two major clades were formed, one containing *D. spectabilis* clustered with *D. punctatus* + *D. tabulaeformis*, and *D. superans* as the sister group to this three-taxon clade, the other containing *D. kikuchii* and *D. houi*. Comparative analyses of the congeneric mitochondrial genomes were performed, which showed that non-coding regions were more variable than the A+T rich region. The mitochondrial nucleotide diversity was more variable when compared within than among genus, and the concatenated tRNA region was the most conserved and the nd6 genes was the most variable.

## INTRODUCTION

Pine moths in the genus *Dendrolimus* (Lasiocampidae) are major economic pests of coniferous trees, such as *Pinus*, *Larix*, *Picea*, and *Abies*. The caterpillars feed extensively on conifer needles; the resulting damage may reduce the tree's seed yield and can lead to heavy defoliation, dieback, and death (*Hou, 1987*; *Chen, 1990*; *Liu & Wu, 2006*; *Zhang et al., 2003a*; *Zhang et al., 2003b*). During an outbreak period, a pine tree can be consumed

Corresponding author
Ai-bing Zhang,
zhangab2008@mail.cnu.edu.cn

in a few days, causing withering and death of pine forests on a large-scale. Furthermore, direct contact with living or dead caterpillars, even their pupae, results in poisoning known as caterpillar arthritis, with serious consequences for human health (*Hou, 1987*). About 30 species of *Dendrolimus* are known to occur in Eurasia, six of them (*D. houi*, *D. kikuchii*, *D. punctatus*, *D. spectabilis*, *D. superans*, *D. tabulaeformis*) are widely distributed in China (*Hou, 1987*; *Chen, 1990*). Meanwhile, *D. pini* was widely distributed across Europe, Central Asia, and North Africa and *D. sibiricus* was mainly detected in western Russia (*Diaz, 2005*; *Baranchikov, Pet'ko & Ponomarev, 2006*; *Mikkola & Ståhls, 2008*). These species are closely related and their discrimination is challenging (*Zhang et al., 2004*; *Mikkola & Ståhls, 2008*). Taxonomy and phylogenetic relationships of *Dendrolimus* were not entirely established, and there are many species with a controversial taxonomic position. Not only that, the co-existence of different species of these pine moths and natural hybrids lead to great difficulties for the species identification. Morphological diagnoses have proven difficult because many of the characters commonly used to distinguish pine moth species are non-discrete and overlapping amongst the species. Furthermore, some *Dendrolimus* species are sympatric coexistence and sharing similar host plants (*Tsai & Liu, 1962*). Hybridization experiments and several molecular studies have been conducted, but no consensus has been achieved regarding their species status (*Zhao et al., 1992*; *Mikkola & Ståhls, 2008*; *Dai et al., 2012*; *Zhang et al., 2014*; *Kononov et al., 2016*).

Mitochondrial genomes (mitogenomes) have been widely used in phylogenetic, population genetics and comparative genomics studies (*Wilson et al., 2000*; *Simon et al., 2006*; *Salvato et al., 2008*; *Cameron, 2014*; *Qin et al., 2015*). Insect mitogenomes have relatively stable structure, such as double-stranded, circular DNA molecule, 14–20 kb in size, comprising 37 genes including 13 protein-coding genes (PCGs) (*Boore, 1999*). Due to its nature of maternal inheritance, mitogenomes has a fast rate of evolution and is particularly useful in phylogenetic analysis (*Hebert, Cywinska & Ball, 2003*). In addition, whole mitogenome sequences can also provide sets of genome-level characters, such as the relative position of different genes, structural genomic features and compositional features, which could be quite useful in phylogenetic analysis (*Thao, Baumann & Baumann, 2004*; *Masta & Boore, 2008*).

Whole mitogenomes instead of several separated gene fragments have been used extensively to construct phylogenies, which providing higher support levels (*Boore, 2006*; *Yang et al., 2015*). Within the order lepidoptera, multiple studies have used mitogenomes to reconstruct the phylogenetic relationships among and within superfamilies (*Whiting et al., 1997*; *Yang et al., 2009*; *Timmermans, Lees & Simonsen, 2014*). Technological advancements have triggered rapid increases in the amount of whole mitogenomes, up to 500 of insect mitogenome have been deposited in GenBank (*Timmermans, Lees & Simonsen, 2014*). However, one of the most recent report shows that only 140 complete lepidoptera mitogenomes (28 families from 12 superfamilies) have been sequenced, and only 64 are available for moth species (*Ramírez-Ríos et al., 2016*). The ease and decreased cost of obtaining whole mitogenome sequences has provided the possibility of comparative genomic studies across short evolutionary distances (i.e., congeneric)

**Table 1 Sample localities in China, geographical coordinates and altitude of *Dendrolimus superans*, *D. kikuchii*, and *D. houi*.**

| Specimen (code) | Sample locality | Latitude (°N) | Longitude (°E) | Altitude (m) |
|---|---|---|---|---|
| *D. superans*04 (LY04) | Keshenketengqi, Inner Mongolia | 43.15 | 117.32 | 1,198 |
| *D. superans*08 (LY08) | Keshenketengqi, Inner Mongolia | 43.15 | 117.32 | 1,198 |
| *D. kikuchii*12 (SM12) | Shilin, Yunnan | 24.45 | 103.16 | 1,888 |
| *D. kikuchii*22 (SM22) | Zhenyuan, Guizhou | 23.43 | 101.08 | 1,057 |
| *D. houi*05 (YN05) | Zhenyuan, Guizhou | 23.43 | 101.08 | 1,057 |
| *D. houi*11 (YN11) | Jingdong, Yunnan | 24.27 | 100.05 | 1,444 |

(*Curole & Kocher, 1999*) providing an understanding of evolutionary dynamics and trends in a phylogenetic framework.

In this study, six complete mitogenomes from three species (*D. superans*, *D. houi*, and *D. kikuchii*, two individuals per species) were newly sequenced. These were combined with the complete mitogenomes of three other species (*D. punctatus*, *D. tabulaeformis*, *D. spectabilis*), which have been published previously (*Qin et al., 2015*), to investigate the taxonomic status of species in the genus *Dendrolimus*. To place the relationships of the genus *Dendrolimus* within a broader context, we also conducted phylogenetic analyses of mitogenomes from other lepidopteran species (mainly moth species). In order to investigate the evolutionary dynamics among six *Dendrolimus* species, comparative analyses were conducted based on 14 mitogenomes (including two subspecies of *D. punctatus*), comparing nucleotide composition, codon usage, differences of overlap and non-coding regions.

## MATERIALS AND METHODS

### Sample collection, DNA extraction, PCR amplification, sequencing, sequence assembly, and annotation

Adult pine moth specimens were sampled at four locations in China, including Inner Mongolia, Yunnan and Guizhou provinces (Table 1). All specimens were preserved in 95% ethanol in the field and stored at 4 °C in the laboratory until DNA extraction. The specimens were identified by Chun-sheng Wu, Institute of Zoology, Chinese Academy of Sciences, China, using morphological characters. Six individuals of three species (*D. kikuchii*, *D. houi*, and *D. superans*, two individuals for each species) were selected for sequencing in this study. Total genomic DNA was extracted from thoracic muscle tissue and leg muscle tissue using a DNeasy BLOOD and Tissue kit (QIAGEN, Hilden, Germany) following the manufacturer's protocol.

Mitochondrial genomes were PCR-amplified and sequenced as described in our previous study (*Qin et al., 2015*). In brief, universal primers were used to amplify initial fragments. Specific fragments were then designed to amplify overlapping regions (i.e., primer walking) (*Salvato et al., 2008*; *Gissi, Iannelli & Pesole, 2008*). PCR recipes and conditions followed *Qin et al. (2015)*. All reactions were performed using Takara LA taq

(TaKaRa Co., Dalian, China). PCR fragments containing the control region were cloned into the pEASY-T3 Cloning Vector (Beijing TransGen Biotech Co., Ltd., Beijing, China) and then sequenced by using tailed primers, M13-F (CGCCAGGGTTTTCCCAGTCACGAC) and M13-R (GAGCGGA TAACAATTT CACACAGG) primers.

Raw sequences were checked manually and assembled on the basis of overlapping regions with the Bioedit V7.0.5 (*Hall, 1999*). The tRNA genes were identified by tRNAscan-SE Search Server v.1.21 (*Simon et al., 1994*). Protein-coding and rRNA genes were determined by comparing homologous sequences with other published lepidoptera mitochondrial genomes (following (*Qin et al., 2015*)). The sequence data have been deposited in GenBank under accession numbers KY000409–KY000414.

## Phylogenetic analysis

To place the relationships of the genus *Dendrolimus* within a broader context, we reconstructed the phylogenetic tree of mitogenomes from other 64 lepidopteran species (mainly moth species). On the other hand, in order to investigate the phylogenetic status of Lasiocampoidea, comparative analyses were conducted based on 14 mitogenomes (including six species) of Lasiocampoidea and 13 mitogenomes of Bombycoidea. Other than the six mitogenomes obtained in this study, other complete mitogenomes were mined from Genbank representing lepidopteran species from 10 superfamilies (Supplemental Information 1). Moreover, in order to clarify the relationship between *D. superans* and *D. sibiricus*, the *COI* and *COII* sequences of *D. sibiricus* (GenBank No. KJ007773, KJ007800, KJ007771, and KJ007815) were used as supplement data to reconstructed phylogenetic tree, which was carried out based on 13 PCGs with maximum likelihood (ML) method.

Nucleotide sequences of the 13 PCGs were aligned based on the translated amino acid sequences using a customized perl script. Non-protein coding region were aligned using MUSCLE with default settings (*Edgar, 2004*). The separated genes and partitions were concatenated with SequenceMatrix software (*Vaidya, Lohman & Meier, 2011*). The concatenated sets of nucleotides were organized into two datasets: dataset 1 representing the 13 PCGs only and dataset 2 representing 37 genes (13 PCGs + 22 transfer RNA genes (tRNA) + 2 ribosomal RNA genes (rRNA)). Substitution saturations of 2 datasets were tested with software DAMBE (*Xia & Xie, 2001*), and both datasets were used in phylogenetic analyses, under the optimality criteria of ML and Bayesian inference (BI) (*Ronquist & Huelsenbeck, 2003*).

In order to standardize the partitioning strategy as recommended for phylogenetic analyses with mitogenomes (*Zardoya & Meyer, 1996*), PartitionFinder v1.1.1 software was used to select the optimal partitioning scheme and to find the best-fitting substitution model for each partition under the Bayesian Information Criterion (*Lanfear et al., 2012*). Not only that, optimized nucleotide substitution models could avoid being affected by the long branch attraction to some extent (*Bergsten, 2005*). The maximum possible partition scheme was 15 partitions: each PCG as a separate partition, the concatenated 22 tRNA genes and the concatenated rRNA genes).

Maximum likelihood analysis was performed with RAxML v7.9.6 and BI analysis with a parallel version of MrBayes v 3.2.2 (*Stamatakis, 2006*; *Ronquist et al., 2012*). The GTR+G+I model was selected for each partition in the two datasets. Support values for the ML topologies were evaluated via bootstrap tests with 1,000 iterations (in RaxML). BI analysis was conducted with two sets of four independent Markov chains run for 10 million Metropolis-coupled generations, with tree sampling occurring every 1,000 generations, and burn-in set to 25% of the trees. After 10 million generations, all runs reached stationary as determined by the program Tracer v1.5.0 (*Rambaut & Drummond, 2007*).

## Genetic distance analysis among closely related species of *Dendrolimus*

In order to test the intraspecific and interspecific differentiation of *Dendrolimus*, 14 mitogenomic were used to calculate the genetic distance across the two datasets described above, which including two subspecies of *D. punctatus* (*D. punctatus punctatus* and *D. punctatus wenshanensis)* and other five species. Genetic distances were calculated using the GTR model selected as the best model by akaike information criterion which performed with Modeltest 3.7 (*Posada & Buckley, 2004*; *Ronquist et al., 2012*). Genetic distances were calculated using a custom C++ script that uses the bio++ function library (*Guéguen et al., 2013*). A correlation matrix was also estimated according to obtained genetic distance matrix. Correlation values ranged from −1 to 1, where values closer to one are indicative of a closer relationship. A graphical visualization of the genetic distances and correlation matrix was drawn using the corrplot.mixed function in R package (*Wei, 2013*).

## Comparative mitogenome analyses of *Dendrolimus*

Nucleotide composition, codon usage (excluding stop codons) and relative synonymous codon usage (RSCU) were calculated across 14 mitogenomes of *Dendrolimus* with MEGA 5.0 (*Tamura, 2011*). Composition skew was calculated using the formulae: AT skew $= (A−T)/(A+T)$ and GC skew $= (G−C)/(G+C)$ (*Perna & Kocher, 1995*). Sliding window analyses were used to calculate nucleotide diversity values across PCGs and regions, which executed with DnaSP software (*Librado & Rozas, 2009*). The window size and step size were set to 100 and 25bp, separately.

# RESULTS

## Phylogenetic analyses of Lasiocampoidea

In the phylogenetic analyses of 78 moth mitogenomes, the monophyly of each superfamily was generally well-supported, in which Lasiocampoidea and Bombycoidea were monophyletic and clustered together as sister groups with high support (Figs. 1 and 2). Similar trees were obtained based on both two datasets (13PCGs and 37 genes), the only difference was among the superfamilies Bombycoidea, Geometroidea, Lasiocampoidea and Noctuoidea, which altogether constitute approximately 73,000 described species (*Minet, 1991*). The 13 PCGs dataset phylogeny placed Geometroidea with Bombycoidea

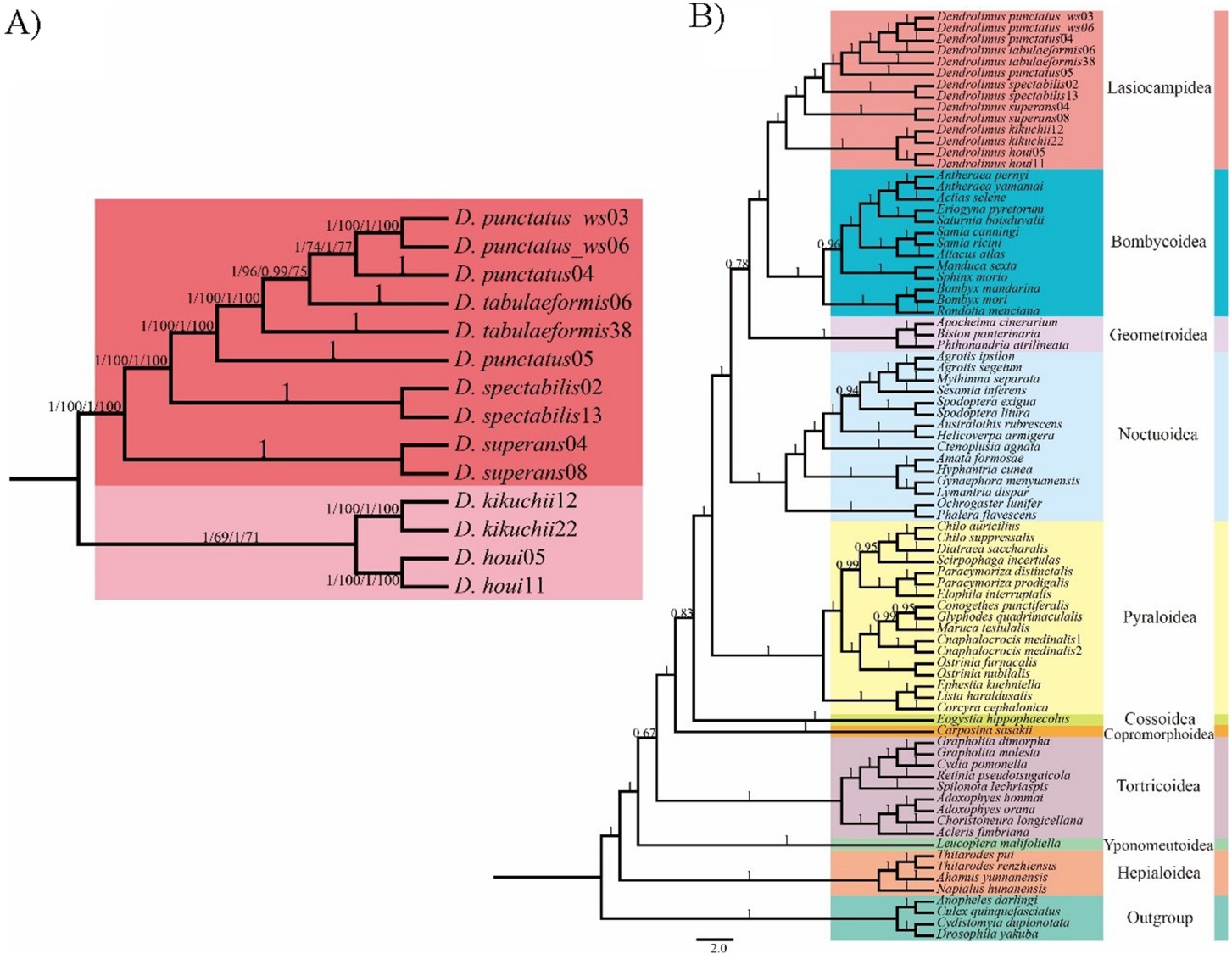

**Figure 1 Phylogenetic relationship of *Dendrolimus*.** (A) Cladogram (ML and BI) depicting six *Dendrolimus* species constructed with Maximum Likelihood and Bayesian inference analyses of (i) 13 protein coding genes (13PCGs); (ii) 37 genes (13 protein-coding genes + 22 transfer RNA genes + 2 ribosomal RNA genes, 37 gene). Numbers above or below branches indicate posterior probabilities and bootstrap percentages across the difference analyses and datasets (13PCGs-BI/13PCGs-ML/37gene-BI/37gene-ML). (B) Cladogram constructed using Bayesian inference analysis of nucleotide sequences of 13 mitochondrial protein-coding genes of Lepidopteran (moth) species, plus outgroups. Numbers above or below branches indicate posterior probabilities.

and Lasiocampoidea, and Noctuoidea as the sister group to this three-taxon clade (53% BP support and 0.78 posterior probabilities) (Fig. 1), which revealed similar relationship with a prior study (*Van Nieukerken et al., 2011*). Nonetheless, the 37 gene dataset phylogeny (Supplemental Information 2) placed Bombycoidea + Lasiocampoidea as the sister group to Geometroidea + Noctuoidea with higher branch support (100% BP support and 1.0 posterior probabilities). The latter relationship was demonstrated with morphological and multigenetic proofs (*Regier et al., 2009*; *Nieukerken Van et al., 2011*; *Bazinet et al., 2013*; *Kawahara & Breinholt, 2014*).

A)

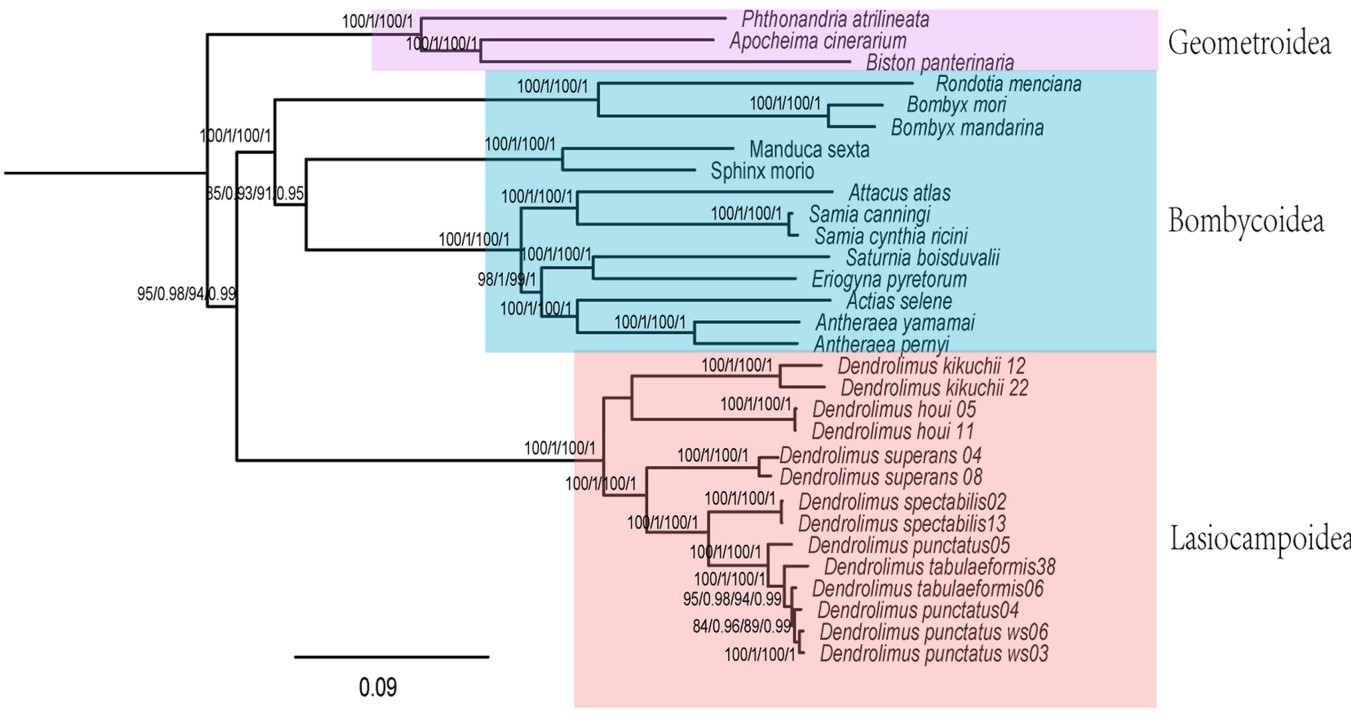

B)

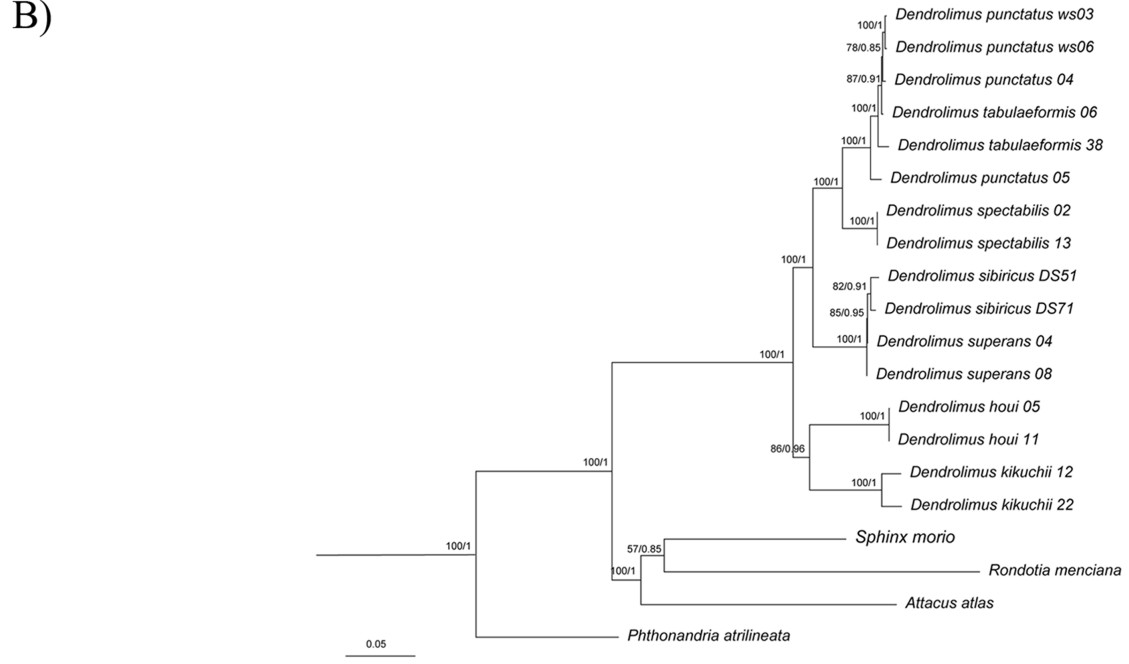

**Figure 2 Phylogeny of Lasiocampoidea.** (A) Phylogenetic tree (ML and BI)of Bombycoidea + Lasiocampoidea species with outgroups (Geoetroidea) constructed with (1) 13 protein coding genes(13PCGs); (2) 37 genes (13 protein-coding genes + 22 transfer RNA genes + 2 ribosomal RNA genes, bootstrap, 37 genes) Numbers above branches indicate bootstrap values, in the order of presentation 13PCGs-ML/13 PCGs-BI/37gene-ML/37gene-BI. Clades with different colors indicate different superfamilies. (B) Phylogeny of *Dendrolims* based on 13 protein coding genes (13PCGs). Only *COI* and *COII* sequences of *D. sibircus* were used. Numbers above or below branches indicate bootstrap values, in the order of presentation 13PCGs-ML/13PCGs-BI.

## Phylogenetic and genetic distance analyses of *Dendrolimus*

Phylogenetic analyses of *Dendrolimus* resulted in a fully resolved tree with robust support for nearly all nodes (Figs. 1 and 2). Phylogenetic analyses inferred from different datasets exhibited the same topology. Six species formed two major clades: *D. punctatus* + *D. tabulaeformis* + *D. spectabilis* + *D. superans* (Clade 1) was the sister group to *D. kikuchii* and *D. houi* (Clade 2). Within "Clade 1," *D. spectabilis* clustered with *D. punctatus* + *D. tabulaeformis*, and *D. superans* was the sister group to this three-taxon clade (Fig. 1A). Among six morphospecies, *D. superans* was successfully found as monophyletic species of these three species (*D. tabulaeformis*, *D. punctatus*, and *D. spectabilis*), which was also confirmed by the results based on *COI*, *COII* and ribosomal internal transcribed spacer (*ITS*) (*Dai et al., 2012*; *Kononov et al., 2016*). Different from the previous view that *D. sibiricus* was a subspecies of *D. superans* (Liu & Wu, 2016), *Kononov et al. (2016)* demonstrate that *D. sibiricus* and *D. superans* were clearly distinguished from each other based on the phylogenetic analysis of *ITS2* sequences. According to the phylogenetic analyses inferred from *COI* and *COII* sequences, *D. sibiricus* and *D. superans* were closed related species (Fig. 1B). However, the taxonomic status of *D. sibiricus* was still ambiguous and more data from nuclear markers were required.

The genetic distance analyses produced results which were consistent with the results of the phylogenetic analyses. The correlation values obtained from genetic distance analysis among specimens of *Dendrolimus* showed that in many cases intraspecific and interspecific values were very similar. Values for intraspecific and interspecific correlations in the group comprising *D. tabulaeformis* and two subspecies of *D. punctatus* were equal or very close to one, which suggests these sequences all have quite a few differences, which would generally be regarded within the range of intraspecific variation. To illustrate the relationship of *Dendrolimus* more clearly, we re-calculated genetic distance with considered *D. punctatus* and *D. tabulaeformis* as an integral taxon (Group A). The genetic distance between *D. spectabilis* and Group A were 0.05, whereas *D. superans* and Group A were 0.07 (Fig. 2). Furthermore, both the correlation value between *D. houi*—Group A and *D. kikuchii*—Group A were negative, highlighting the relatively distant genetic relationship with other four species (*D. punctatus*, *D. tabulaeformis*, *D. spectabilis*, and *D. superans*) (Fig. 3).

## Comparative mitochondrial genome characterization of *Dendrolimus*

### *Mitochondrial genome organization*

The complete mitochondrial genomes of *Dendromlius* ranged from 15,370 to 15,417 bp in length (Table 2). The gene order was identical to other ditrysian lepidopterans with the standard *trnM* gene location type (*trnM-trnI-trnQ*), and all mitochondrial genomes exhibit similar sequence characteristics. The mitochondrial genes of three newly-sequenced *Dendrolimus* species (*D. superans*, *D. houi*, and *D. kikuchii*) are coded on the majority strand, except for four PCG (*nd5*, *nd4*, *nd4L*, and *nd1*) and eight tRNA genes (*trnQ*, *trnC*, *trnY*, *trnF*, *trnH*, *trnP*, *trnL(CUN)*, and *trnV*) (Table 2).

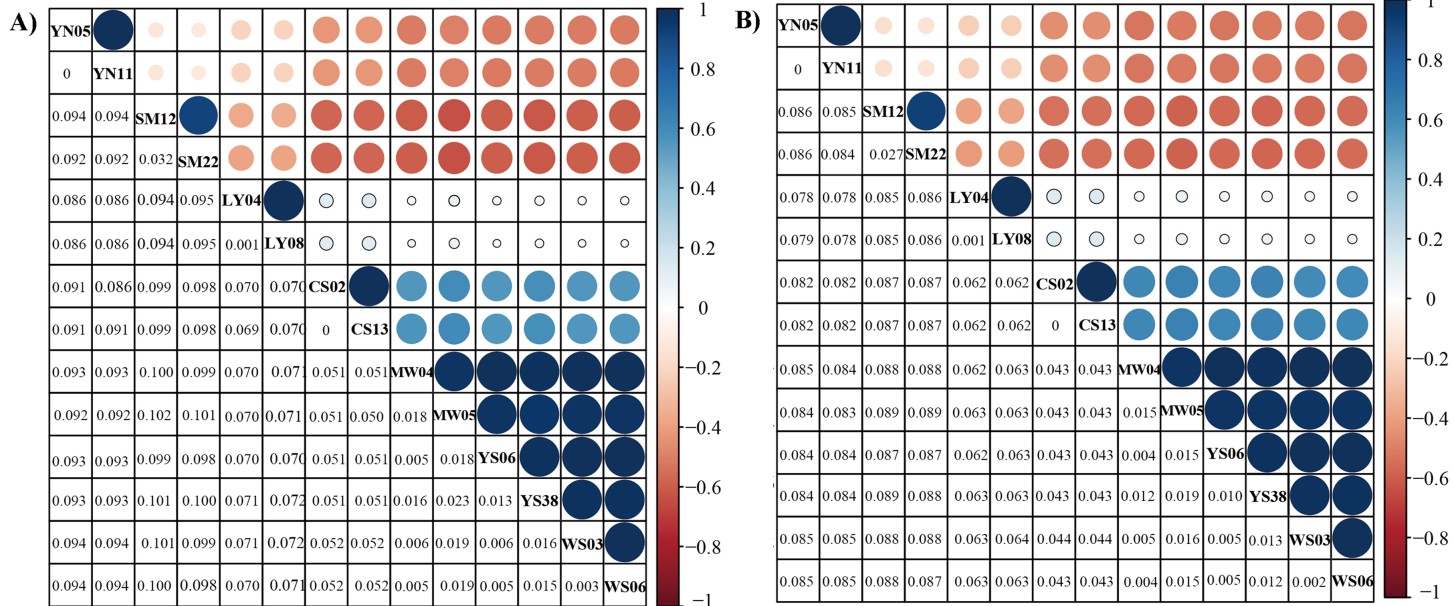

**Figure 3 Genetic distance of six *Dendrolimus* species.** Genetic distance (below diagonal) and correlation relationship (above diagonal) of (A) 13 concatenate protein coding genes and (B) 37 concatenate genes. The size of circle stands for the correlation values, which range from −1 to 1. Values closer to one indicate a closer relationship. Species names were abbreviated: *D. spectabilis* (CS02 and CS13), *D. tabulaeformis* (YS06 and YS08), *D. punctatus punctatus* (MW04 and MW05), *D. punctatus wenshanensis* (WS03 and WS06), *D. superans* (LY04 and LY08), *D. kikuchii* (SM12 and SM22), *D. houi* (YN05 and YN11).

### Base composition and skewness

Metazoan mitogenomes usually exhibit a clear strand bias toward adenine (A) and thymine (T) in nucleotide composition. Consistent with previous observations of *Dendrolimus* mitogenomes, the mitochondrial sequence of three newly-sequenced *Dendrolimus* species were biased toward A and T. The A+T content of the majority strand ranged from 78.7% and 78.8% for *D. kikuchii*, 80% and 79.9% for *D. houi*, and 80.1% and 80.2% for *D. superans* (Supplemental Information 3). The strand bias also can be measured as AT- and GC-skews. The average AT-skew across all available *Dendrolimus* mitochondrial genomes was 0.028, ranging from 0.037 to 0.017, whereas the average GC-skew of the *Dendrolimus* mitochondrial genomes was −0.23, ranging from −0.26 to −0.22.

### Start and stop codon usage

Start and stop codon usage is an important characteristic in the annotation of PCGs. We compared the start and stop codons across the six species of *Dendrolimus* (Table 3). All PCGs started with the typical ATN codons except for *cox1* which used CGA. Most of the start codon were consistent within the six species but a few were different (*nd2*, *cox2*, *atp8*, *nd3*, *nd5*, *nd1*). This was especially the case for *atp8* and *nad3*, which were the most variable among the genes. It is noteworthy that *atp8* and *nad3* are the shortest PCGs when compare to others in the mitochondrial genome, suggesting variability in start codon usage maybe related to gene length.

**Table 2 Genome organization of *D. kikuchii*, *D. houi*, and *D. superans*.**

| Gene | Strand | Location | | | | | |
|------|--------|----------|----------|----------|----------|----------|----------|
| | | SM12 | SM22 | YN05 | YN11 | LY04 | LY08 |
| trnM | F | 1–67 | 1–67 | 1–67 | 1–67 | 1–68 | 1–67 |
| trnI | F | 71–134 | 71–134 | 69–132 | 69–132 | 72–135 | 72–135 |
| trnQ | R | 132–200 | 132–200 | 130–198 | 130–198 | 133–201 | 133–201 |
| nad2 | F | 257–1,264 | 255–1,262 | 254–1,264 | 254–1,264 | 256–1,266 | 256–1,266 |
| trnW | F | 1,264–1,334 | 1,262–1,332 | 1,263–1,332 | 1,263–1,332 | 1,265–1,336 | 1,265–1,335 |
| trnC | R | 1,327–1,392 | 1,325–1,390 | 1,325–1,391 | 1,325–1,391 | 1,329–1,394 | 1,328–1,393 |
| trnY | R | 1,393–1,460 | 1,391–1,458 | 1,393–1,463 | 1,393–1,463 | 1,395–1,460 | 1,394–1,459 |
| cox1 | F | 1,501–3,031 | 1,500–3,030 | 1,492–3,022 | 1,492–3,022 | 1,493–3,023 | 1,492–3,022 |
| trnL(UUR) | F | 3,032–3,098 | 3,031–3,097 | 3,023–3,089 | 3,023–3,089 | 3,024–3,090 | 3,023–3,089 |
| cox2 | F | 3,099–3,780 | 3,098–3,779 | 3,090–3,773 | 3,090–3,773 | 3,091–3,772 | 3,090–3,771 |
| trnK | F | 3,781–3,851 | 3,780–3,850 | 3,775–3,845 | 3,775–3,845 | 3,773–3,843 | 3,772–3,842 |
| trnD | F | 3,852–3,919 | 3,851–3,918 | 3,847–3,913 | 3,847–3,913 | 3,847–3,914 | 3,846–3,913 |
| atp8 | F | 3,920–4,081 | 3,919–4,080 | 3,914–4,075 | 3,914–4,075 | 3,915–4,076 | 3,914–4,075 |
| atp6 | F | 4,075–4,752 | 4,074–4,751 | 4,069–4,746 | 4,069–4,746 | 4,070–4,747 | 4,069–4,746 |
| cox3 | F | 4,759–5,547 | 4,760–5,548 | 4,762–5,550 | 4,762–5,550 | 4,760–5,548 | 4,759–5,547 |
| trnG | F | 5,550–5,616 | 5,551–5,617 | 5,553–5,618 | 5,553–5,618 | 5,551–5,616 | 5,550–5,615 |
| nad3 | F | 5,617–5,970 | 5,618–5,971 | 5,619–5,970 | 5,619–5,970 | 5,617–5,970 | 5,616–5,969 |
| trnA | F | 5,973–6,039 | 5,974–6,039 | 5,971–6,038 | 5,971–6,038 | 5,975–6,041 | 5,974–6,040 |
| trnR | F | 6,058–6,123 | 6,058–6,123 | 6,048–6,112 | 6,048–6,112 | 6,055–6,118 | 6,054–6,117 |
| trnN | F | 6,145–6,211 | 6,145–6,211 | 6,118–6,184 | 6,118–6,184 | 6,120–6,185 | 6,119–6,184 |
| trnS(AGN) | F | 6,226–6,293 | 6,219–6,286 | 6,210–6,277 | 6,210–6,277 | 6,202–6,269 | 6,201–6,268 |
| trnE | F | 6,293–6,357 | 6,286–6,350 | 6,277–6,345 | 6,277–6,345 | 6,269–6,333 | 6,268–6,333 |
| trnF | R | 6,371–6,437 | 6,364–6,430 | 6,354–6,420 | 6,354–6,420 | 6,346–6,412 | 6,346–6,412 |
| nad5 | R | 6,442–8,181 | 6,435–8,174 | 6,425–8,167 | 6,425–8,167 | 6,416–8,158 | 6,416–8,158 |
| trnH | R | 8,182–8,246 | 8,175–8,239 | 8,168–8,231 | 8,168–8,231 | 8,159–8,226 | 8,159–8,226 |
| nad4 | R | 8,247–9,585 | 8,240–9,578 | 8,232–9,570 | 8,232–9,570 | 8,227–9,565 | 8,227–9,565 |
| nad4l | R | 9,620–9,913 | 9,611–9,904 | 9,602–9,895 | 9,602–9,895 | 9,604–9,897 | 9,604–9,897 |
| trnT | F | 9,918–9,982 | 9,909–9,973 | 9,900–9,964 | 9,900–9,964 | 9,905–9,970 | 9,905–9,969 |
| trnP | R | 9,983–10,047 | 9,974–10,038 | 9,965–10,029 | 9,965–10,029 | 9,971–10,035 | 9,970–10,034 |
| nad6 | F | 10,056–10,586 | 10,047–10,577 | 10,038–10,568 | 10,038–10,568 | 10,044–10,574 | 10,043–10,573 |
| cob | F | 10,590–11,738 | 10,581–11,729 | 10,574–11,722 | 10,574–11,722 | 10,579–11,727 | 10,578–11,726 |
| trnS(UCN) | F | 11,737–11,802 | 11,728–11,793 | 11,726–11,791 | 11,726–11,791 | 11,731–11,797 | 11,730–11,796 |
| nad1 | R | 11,802–12,755 | 11,793–12,746 | 11,791–12,744 | 11,791–12,744 | 11,797–12,750 | 11,796–12,749 |
| trnL(CUN) | R | 12,757–12,827 | 12,748–12,818 | 12,746–12,813 | 12,746–12,813 | 12,752–12,820 | 12,751–12,819 |
| rrnL | R | 12,828–14,210 | 12,819–14,204 | 12,814–14,220 | 12,814–14,220 | 12,821–14,253 | 12,820–14,253 |
| trnV | R | 14,211–14,275 | 14,205–14,268 | 14,221–14,286 | 14,221–14,286 | 14,254–14,319 | 14,254–14,319 |
| rrnS | R | 14,276–15,058 | 14,269–15,051 | 14,287–15,062 | 14,287–15,063 | 14,320–15,101 | 14,320–15,100 |
| AT region | F | 15,059–15,377 | 15,052–15,370 | 15,063–15,381 | 15,064–15,382 | 15,102–15,417 | 15,101–15,417 |

**Note:**

Genome organization of *D. kikuchii* (sample ID: SM12 and SM22), *D. houi* (sample ID: YN05 and YN11), and *D. superans* (sample ID: LY04 and LY08).

Table 3 Start codon and stop codon of 13 protein coding genes in six *Dendrolimus* species.

| Samples | nad2 | cox1 | cox2 | atp8 | atp6 | cox3 | nad3 | nad5 | nad4 | nad4l | nad6 | cob | nad1 |
|---------|------|------|------|------|------|------|------|------|------|-------|------|-----|------|
| *D. spectabilis*02 | ATT/TAA | CGA/T | ATA/T | ATC/TAA | ATG/TAA | ATG/TAA | ATC/TA | ATT/TAA | ATG/T | ATG/TAA | ATA/TAA | ATG/TAA | ATG/TAA |
| *D. spectabilis*13 | ATT/TAA | CGA/T | ATA/T | ATC/TAA | ATG/TAA | ATG/TAA | ATC/TA | ATT/TAA | ATG/T | ATG/TAA | ATA/TAA | ATG/TAA | ATG/TAA |
| *D. tabulaeformis*06 | ATT/TAA | CGA/T | ATA/T | ATT/TAA | ATG/TAA | ATG/TAA | ATG/TA | ATT/TAA | ATG/T | ATG/TAA | ATA/TAA | ATG/TAA | ATG/TAA |
| *D. tabulaeformis*38 | ATT/TAA | CGA/T | ATA/T | ATC/TAA | ATG/TAA | ATG/TAA | ATT/TA | ATT/TAA | ATG/T | ATG/TAA | ATA/TAA | ATG/TAA | ATG/TAA |
| *D. punctatus*04 | ATT/TAA | CGA/T | ATA/T | ATT/TAA | ATG/TAA | ATG/TAA | ATG/TA | ATT/TAA | ATG/T | ATG/TAA | ATA/TAA | ATG/TAA | ATG/TAA |
| *D. punctatus*05 | ATT/TAA | CGA/T | ATA/T | ATT/TAA | ATG/TAA | ATG/TAA | ATA/TA | ATT/TAA | ATG/T | ATG/TAA | ATA/TAA | ATG/TAA | ATG/TAA |
| *D. punctatus_ws*03 | ATT/TAA | CGA/T | ATA/T | ATT/TAA | ATG/TAA | ATG/TAA | ATG/TA | ATT/TAA | ATG/T | ATG/TAA | ATA/TAA | ATG/TAA | ATG/TAA |
| *D. punctatus_ws*06 | ATT/TAA | CGA/T | ATA/T | ATT/TAA | ATG/TAA | ATG/TAA | ATG/TA | ATT/TAA | ATG/T | ATG/TAA | ATA/TAA | ATG/TAA | ATG/TAA |
| *D. kikuchii*12 | ATT/TAA | CGA/T | ATA/T | ATA/TAA | ATG/TAA | ATG/TAA | ATT/TAA | ATA/TAA | ATG/T | ATG/TAA | ATA/TAA | ATG/TAA | GTG/TAA |
| *D. kikuchii*22 | ATT/TAA | CGA/T | ATA/T | ATA/TAA | ATG/TAA | ATG/TAA | ATT/TAA | ATA/TAA | ATG/T | ATG/TAA | ATA/TAA | ATG/TAA | GTG/TAA |
| *D. houi*05 | ATT/TAA | CGA/T | ATA/TAG | ATT/TAA | ATG/TAA | ATG/TAA | ATT/T | ATT/TAA | ATG/T | ATG/TAA | ATA/TAA | ATG/TAA | ATG/TAA |
| *D. houi*11 | ATT/TAA | CGA/T | ATA/TAG | ATT/TAA | ATG/TAA | ATG/TAA | ATT/T | ATT/TAA | ATG/T | ATG/TAA | ATA/TAA | ATG/TAA | ATG/TAA |
| *D. superans*04 | ATC/TAA | CGA/T | ATA/T | ATT/TAA | ATG/TAA | ATG/TAA | ATC/TAA | ATT/TAA | ATG/T | ATG/TAA | ATA/TAA | ATG/TAA | ATG/TAA |
| *D. superans*08 | ATC/TAA | CGA/T | ATA/T | ATT/TAA | ATG/TAA | ATG/TAA | ATC/TAA | ATT/TAA | ATG/T | ATG/TAA | ATA/TAA | ATG/TAA | ATG/TAA |

Nine genes (*nd2*, *atp8*, *atp6*, *cox3*, *nd5*, *nd4l*, *nd6*, *cob*, *nd1*) share the same complete stop codon TAA, and four genes use incomplete stop codons (*cox1*, *cox2 and nd4*, *nd3*) (Table 3). Incomplete stop codons are common in lepidopteran mitogenomes and are presumed to be completed via post-transcriptional polyadenylation (*Chen et al., 2016*). Changes in stop codon usage among *Dendrolimus* were rarer than changes in start codon usage. Only in the *cox2* and *nad2* genes, did we observe changes in the stop codon used. Therefore, we can conclude that even within congeneric species, start and stop codons are variable in the mitochondrial genome.

### Codon usage and RSCU

Condon usage and RSCU results were compared across all available *Dendrolimus* mitogenomes (Supplemental Information 4). The analysis showed that Leu2 (UUR), Ile, Phe, Met, Asn, Gly, Ser2 (UCN), Tyr are the eight most frequent amino acids and were represented by at least 50 codons per thousand codons. Two codon families, Leu2 and Ile, had at least 100 codons per thousand codons. Leu2, a hydrophobic amino acid, was significantly more frequent than other amino acids, which may relate to the function of chondriosomes in many transmembrane proteins. The rarest used codon family was Cys.

The usage of both twofold and fourfold degenerate codons was biased towards the use of codons with A or T in third position (Supplemental Information 5). Codons which have relatively high G and C content are likely to be abandoned, reflecting a finding across other lepidopteran insects. Examination of the fourteen individual *Dendrolimus* mitogenomes showed that Leu2 (UUA), Ser2 (UCU), Arg (CGA), Ala (GCU), Ser1 (AGA) are the five most frequent relative synonymous codons.

### Non-coding regions, overlapping regions and A+T rich region

All 14 mitogenomes had six overlapping regions and the size ranged from one to eight bp (Table 4). Nucleotide sequence of six mutual overlapping areas were almost identical,

**Table 4 Sequence length of non-coding and overlapping regions between two genes among 14 individuals of *Dendrolimus* species.**

| Location* | D. spectabilis | | D. tabulaeformis | | D. punctatus punctatus | | D. punctatus wenshanensis | | D. superans | | D. kikuchii | | D. houi | |
|---|---|---|---|---|---|---|---|---|---|---|---|---|---|---|
| | CS02 | CS13 | YS06 | YS38 | MW04 | MW05 | WS03 | WS06 | LY04 | LY08 | SM12 | SM22 | YN05 | YN11 |
| trnM-trnI | 3 | 3 | 3 | 3 | 3 | 3 | 3 | 3 | 3 | 4 | 3 | 3 | 1 | 1 |
| trnI-trnQ | −3 | −3 | −3 | −3 | −3 | −3 | −3 | −3 | −3 | −3 | −3 | −3 | −3 | −3 |
| trnQ-nad2 | 58 | 58 | 58 | 58 | 58 | 58 | 58 | 58 | 54 | 54 | 56 | 54 | 55 | 55 |
| nd2-trnW | −2 | −2 | −2 | −2 | −2 | −2 | −2 | −2 | −2 | −2 | −1 | −1 | −2 | −2 |
| trnW-trnC | −8 | −8 | −8 | −8 | −8 | −8 | −8 | −8 | −8 | −8 | −8 | −8 | −8 | −8 |
| trnC-trnY | 0 | 0 | 0 | 0 | 0 | 0 | 0 | 0 | 0 | 0 | 0 | 0 | 1 | 1 |
| trnY-cox1 | 25 | 25 | 34 | 34 | 27 | 34 | 34 | 34 | 32 | 32 | 40 | 41 | 28 | 28 |
| cox2-trnK | 0 | 0 | 0 | 0 | 0 | 0 | 0 | 0 | 0 | 0 | 0 | 0 | 1 | 1 |
| trnK-trnD | 3 | 3 | 3 | 3 | 3 | 3 | 3 | 3 | 3 | 3 | 0 | 0 | 1 | 1 |
| atp8-atp6 | −7 | −7 | −7 | −7 | −7 | −7 | −7 | −7 | −7 | −7 | −7 | −7 | −7 | −7 |
| atp6-cox3 | 11 | 11 | 15 | 15 | 15 | 14 | 15 | 15 | 12 | 12 | 6 | 8 | 15 | 15 |
| cox3-trnG | 2 | 2 | 2 | 2 | 2 | 2 | 2 | 2 | 2 | 2 | 2 | 2 | 2 | 2 |
| nad3-trnA | 0 | 0 | 0 | 0 | 0 | 0 | 0 | 0 | 4 | 4 | 2 | 2 | 0 | 0 |
| trnA-trnR | 20 | 20 | 15 | 15 | 15 | 15 | 15 | 15 | 13 | 13 | 18 | 18 | 9 | 9 |
| trnR-trnN | 4 | 4 | 4 | 4 | 4 | 4 | 4 | 4 | 1 | 1 | 21 | 21 | 5 | 5 |
| trnN-trnS(AGN) | 18 | 18 | 11 | 11 | 11 | 13 | 11 | 11 | 16 | 16 | 14 | 7 | 25 | 25 |
| trnS(AGN)-trnE | −1 | −1 | −1 | −1 | −1 | −1 | −1 | −1 | −1 | −1 | −1 | −1 | −1 | −1 |
| trnE-trnF | 8 | 8 | 4 | 4 | 4 | 4 | 4 | 4 | 12 | 12 | 13 | 13 | 8 | 8 |
| trnF-nad5 | 3 | 3 | 2 | 2 | 2 | 3 | 2 | 2 | 3 | 3 | 4 | 4 | 4 | 4 |
| nad4-nd4l | 23 | 23 | 24 | 24 | 24 | 19 | 24 | 24 | 38 | 38 | 34 | 32 | 31 | 31 |
| nad4l-trnT | 7 | 7 | 7 | 7 | 7 | 7 | 7 | 7 | 7 | 7 | 4 | 4 | 4 | 4 |
| trnP-nad6 | 8 | 8 | 8 | 8 | 8 | 8 | 8 | 8 | 8 | 8 | 8 | 8 | 8 | 8 |
| nad6-cytb | 4 | 4 | 4 | 4 | 4 | 4 | 4 | 4 | 4 | 4 | 3 | 3 | 5 | 5 |
| cytb-trnS(UCN) | 3 | 3 | 3 | 3 | 3 | 3 | 3 | 3 | 3 | 3 | −2 | −2 | 3 | 3 |
| trnS(UCN)-nd1 | −1 | −1 | −1 | −1 | −1 | −1 | −1 | −1 | −1 | −1 | −1 | −1 | −1 | −1 |
| nad1-trnL(CUN) | 1 | 1 | 1 | 1 | 1 | 1 | 1 | 1 | 1 | 1 | 1 | 1 | 1 | 1 |

**Notes:**

Sequence length of non-coding and overlapping regions between two genes among 6 species: D. spectabilis, D. tabulaeformis, D. punctatus, D. superans, D. kikuchii, and D. houi.

Location*: Sequence length between two genes, positive value stands for non-coding regions, negative value stands for overlapping regions.

except for the overlap between *nd2* and *trnW* in *D. kikuchii* which was one bp shorter than other species of *Dendrolimus*. In addition to the control region, there were 17 non-coding regions in the mitogenomes of *D. punctatus*, *D. tabulaeformis* and *D. spectabilis*, 18 in *D. superans*, 16 in *D. kikuchii* and 19 in *D. houi* (Table 4). It is noteworthy that there are 6 intergenic regions, *trnQ-nad2* (54 bp–58 bp), *trnY-cox1* (25–41 bp), *atp6-cox3* (6–15 bp), *trnA-trnR* (9–20 bp), *trnN-trnS* (AGN) (7–25 bp), *nad4-nd4l* (19–38 bp), were longer than 15 bp. To investigate the utility, we constructed a phylogenetic tree of *Dendrolimus* species using only the A+T rich region and intergenic regions (Supplemental Information 5). The phylogenetic analysis using the A+T rich region produced similar but slightly different topology comparing with the whole mitogenomes.

## Sliding-window analysis

Sliding-window analysis was conducted to compare nucleotide diversity among the mitochondrial PCGs and non-coding regions of 14 individuals in *Dendrolimus* (Fig. 4). The intergenic region has the highest nucleotide diversity which is likely attributable to the large indels in this region. This was followed by *nd6*, *cytb*, *cox2*, *atp6*, *cox3*, *nd3*, A+T rich region, *nd1*, *cox1*, *nd2*, *nd5*, *nd4*, *nd4l*, *atp8*, rRNA, tRNA. It is notable that the nucleotide diversity of the A+T rich region was moderate; lower than many PCGs. The tRNA was the most conserved region and *cox1* was the most conserved PCG. In contrast, sliding-window analyses using all 78 lepidopteran mitogenomes (same dataset as the phylogenetic analyses) produced substantially similar patterns: the *nd6* gene had the highest level of divergence and tRNA was the most conserved region, while the *cox1* was the most conserved than all PCGs.

## DISCUSSION

The monophyly of each superfamily was generally well-supported based on the 78 mitogenome analysis, which was consistency with prior studies (*Yang et al., 2009*; *Kawahara & Breinholt, 2014*; *Qin et al., 2015*). Previous studies have included Lasiocampidae within Bombycoidea (*Brock, 1971*; *Scoble, 1992*; *Kawahara & Breinholt, 2014*), while other studies have treated Lasiocampidae as a distinct superfamily Lasiocampoidea (*Minet, 1991*; *Regier et al., 2009*; *Van Nieukerken et al., 2011*; *Zwick et al., 2011*; *Bazinet et al., 2013*; *Wu et al., 2016*). However, according to our results based on mitogenomes, Lasiocampoidea, and Bombycoidea were monophyletic and clustered together as sister groups with high support. Within the Bombycoidea, the relationship among the families Bombycidae, Sphingidae, and Saturniidae has been difficult to resolve in previous study (*Regier et al., 2013*). In our study, the analysis of both datasets placed the Bombycidae as the sister group to Saturniidae and Sphingidae with high support (100% bootstrap), which is consistent with the phylogenetic relationship based on transcriptomic data (2,696 genes) (*Breinholt & Kawahara, 2013*).

The topology of our mitogenome *Dendrolimus* phylogeny showed some differences from the topology proposed by previous studies. *Zhang et al. (2014)* constructed a phylogeny of *Dendrolimus* based on one pheromone-binding proteins (*PBP1*) and two general odorant-binding proteins (*OBPs*) in which *D. kikuchii* and *D. houi* was proposed as basal species of *D. tabulaeformis*, *D. punctatus*, and *D. spectabilis*. The relationships of these three species were verified with mitogenomes analysis, sharing a closer relationship to each other with respect to *D. superans*. This result was also proved by the phylogenetic analysis of *Dendrolimus* based on *COI* and *COII* genes (*Dai et al., 2012*; *Qin et al., 2015*; *Kononov et al., 2016*). Two species, *D. kikuchii* and *D. houi*, were the most basal species in *Dendrolimus*. According to the phylogenetic tree based on whole mitogenomes, these two species could both be accept as monophyletic taxa, which was consistent with previous studies depended on *COII* and *ITS* genes (*Dai et al., 2012*; *Kononov et al., 2016*). This result was different from the previous studies based on the analysis with *PBP1* and *OBPs*. The primary cause of this discrepancy is due to the different modes of inheritance, maternal for mtDNA and biparental for *PBPs*, *OBPs*, and *ITS*. Moreover, the

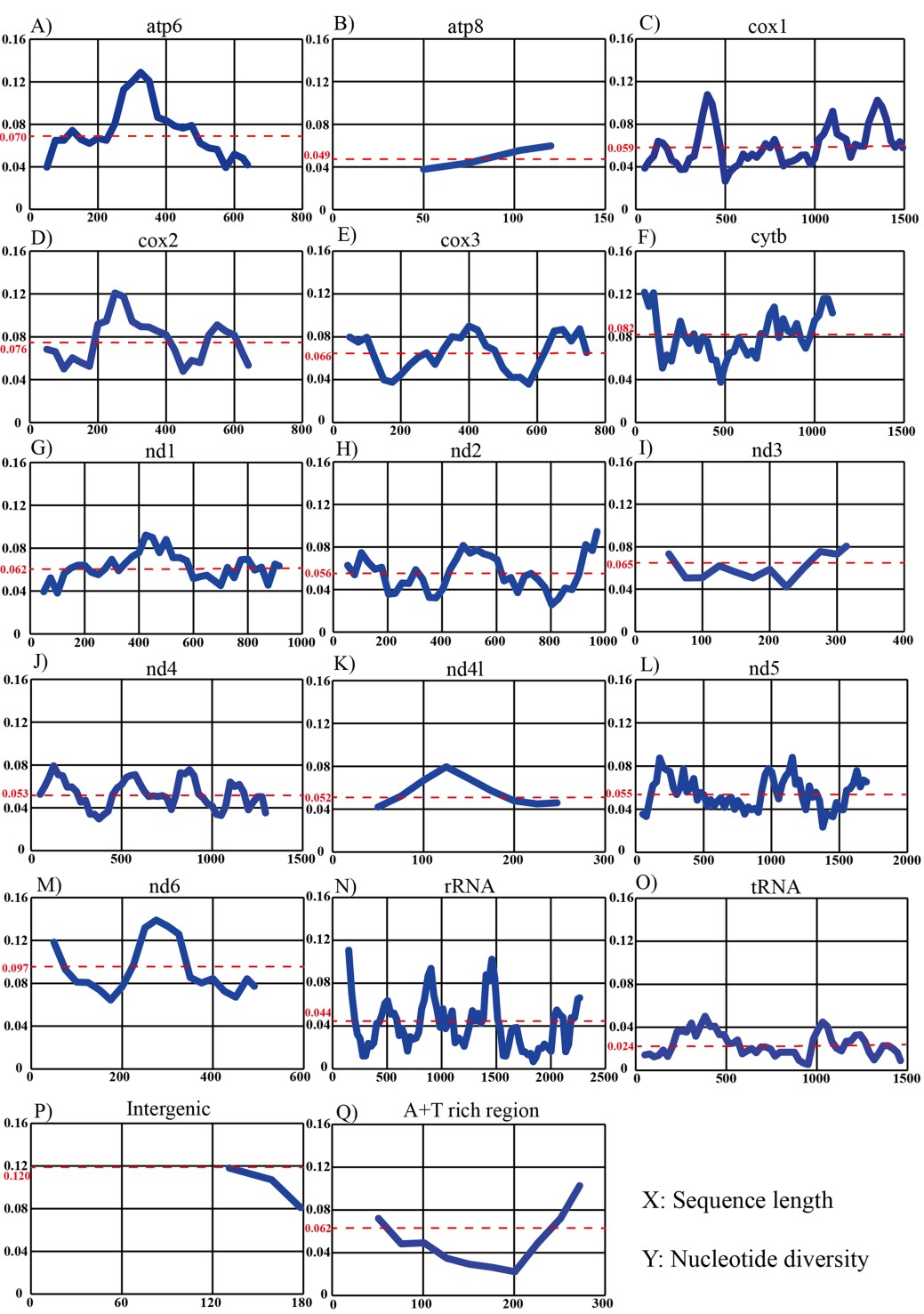

**Figure 4 Sliding-window analyses of mitogenomes among six *Dendrolimus* species.** Sliding-window analyses of 13 protein coding genes (A–M), concatenated tRNA (N) and rRNA genes (O), intergenic (P) and A+T rich region (Q) among six Dendrolimus species. The X-axis represents sequence length, the Y-axis represents nucleotide diversity. The red dotted line indicates the average nucleotide diversity.

dissimilarity in mutation rates and number of information sites could also lead to inconsistent results. For example, *D. kikuchii* and *D. houi* were the most closed related species based on the *COI* gene (*Dai et al., 2012*), however, paraphyletic groups were showed based on 5′-end portion of *COI* gene (*Kononov et al., 2016*). Not only that, the closed relationship of *D. tabulaeformis*, *D. punctatus*, and *D. spectabilis* was also proved by the genetic distance analysis. *D. superans* was the sister species to these three species, which also had different mitogenome component characteristics.

The largest intergenic spacer of whole mitogenome is the A+T rich region, which not only has the characteristics of non-coding genes, but also contains important sites for the regulation of transcription and replication (*Gissi, Iannelli & Pesole, 2008*), as well as useful phylogenetic signals, particularly for determining congeneric relationships and relationships among recently diverged species. The results of phylogenetic analysis using the A+T rich region produced similar but slightly different topology comparing with the whole mitogenomes. This suggests the intergenic regions might be too variable to be useful for phylogenetic analyses; nevertheless, the A+T rich region might be an effective molecular tool in solving phylogenetic relationships among recently diverged species.

## CONCLUSION

In this study, both phylogenetic and genetic distance analyses obtained consistent results regarding the relationships among six closely related species. The whole mitogenomes failed to provide enough information to distinguish *D. tabulaeformis* from *D. punctatus*, which suggest there might not be a clear species boundary between these two species. This finding is consistent with the results of previous studies, in which *D. tabulaeformis* was regarded as ecological type of *D. punctatus* based on several DNA markers and experiments of interspecific hybridization. Meanwhile, *D. spectabilis* fell as sister to these two sibling species, and *D. superans* fell as sister to these three taxa. *D. kikuchii* and *D. houi* are sister species, having relatively close relationship comparing with other four species. The phylogenetic relationships of *Dendrolimus* based on complete mitogenome could provide a theoretical basis for pest control of pine caterpillar, thereby reducing the economic losses of forests.

Congeneric species exhibit similar mitochondrial genome features, such as genome organization, nucleotide composition, codon usage and RSCU. Within the genus *Dendrolimus*, start and stop codons were variable in mitochondrial genome and the change of stop codons were rarer than start codons. Non-coding regions were the most variable regions in mitochondrial genomes. When comparing nucleotide diversity, the *nad6* gene had the highest level of divergence and the tRNA region was the most conserved.

## ACKNOWLEDGEMENTS

The authors wish to thank Prof. Xiang-Bo Kong (Chinese Academy of Forestry) for helping with sample collection.

### Funding

This work was supported by the National Natural Science Foundation of China (Grant No. 31425023, 31400191, 31601877 and 31772501) and the Beijing Municipal Natural Science Foundation (Grant No. 5172005). The funders had no role in study design, data collection and analysis, decision to publish, or preparation of the manuscript.

### Grant Disclosures

The following grant information was disclosed by the authors:
National Natural Science Foundation of China: 31425023, 31400191, 31601877 and 31772501.
Beijing Municipal Natural Science Foundation: 5172005.

### Competing Interests

The authors declare that they have no competing interests.

### Author Contributions

- Jie Qin conceived and designed the experiments, performed the experiments, analyzed the data, contributed reagents/materials/analysis tools, prepared figures and/or tables, authored or reviewed drafts of the paper, approved the final draft.
- Jing Li conceived and designed the experiments, analyzed the data, contributed reagents/materials/analysis tools, prepared figures and/or tables, authored or reviewed drafts of the paper, approved the final draft.
- Qiang Gao contributed reagents/materials/analysis tools, approved the final draft.
- John-James Wilson prepared figures and/or tables, authored or reviewed drafts of the paper, approved the final draft.
- Ai-bing Zhang conceived and designed the experiments, contributed reagents/materials/analysis tools, prepared figures and/or tables, authored or reviewed drafts of the paper, approved the final draft.

### Data Availability

The sequence data are available at GenBank under accession numbers KY000409–KY000414.

### Supplemental Information

Supplemental information for this article can be found online at http://dx.doi.org/10.7717/peerj.7317#supplemental-information.

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
