# Peer review of "Mitochondrial phylogeny and comparative mitogenomics of closely related pine moth pests (Lepidoptera: Dendrolimus)"

_PeerJ, doi:10.7717/peerj.7317_

## Round 0.1 · original submission · Major Revisions

Dear Dr. Qin and colleagues:

Thanks for submitting your manuscript to PeerJ. I have now received three independent reviews of your work, and as you will see, the reviewers raised some concerns about the research. Despite this, these reviewers are optimistic about your work and the potential impact it will lend to research on pine moth mitogenomics and systematics. Thus, I encourage you to revise your manuscript accordingly, taking into account all of the concerns raised by the reviewers.

While the concerns of the reviewers are relatively minor, this is a major revision to ensure that the original reviewers have a chance to evaluate your responses to their concerns.

Please note that reviewer 2 provided edits on marked-up versions of the manuscript. Please address all of these criticisms, as well as those provided in the general reviews.

I look forward to seeing your revision, and thanks again for submitting your work to PeerJ.

Good luck with your revision,

-joe

·

Basic reporting

Clear and unambiguous, professional English used throughout.
Literature references, sufficient field background/context provided.
Professional article structure, figures, tables. Raw data shared.
Self-contained with relevant results to hypotheses.

Experimental design

Original primary research within Aims and Scope of the journal.
Research question well defined, relevant & meaningful. It is stated how research fills an identified knowledge gap.
Rigorous investigation performed to a high technical & ethical standard.
Methods described with sufficient detail & information to replicate.

Validity of the findings

Data is robust, statistically sound, & controlled.
Conclusions are well stated, linked to original research question.

Additional comments

Revision suggestions:
1.concerning the species of Dendrolimus, Fauna Sinica Vol.47(2006) should also be referenced.
2. Six individuals of three species (D. kikuchii, D. houi and D. superans, 2 individuals for each species) were selected for sequencing in this study. Please state more clearly how did you collected 2 individuals, from one site or different sites?
3.It should be stated more clearly that from Mitochondrial phylogeny and comparative mitogenomics of closely related pine moth pests what results were different from the results of previous studies. And should also discuss the reasons.

·

Basic reporting

This is an interesting ms based on a large dataset of mitochondrial genomes of an important group of pine defoliating pests in the Palaearctic region and in Asia.

I have two major issues:

1. an important species, actually the species with the largest range in Asia and occurring also in China, Dendrolimus sibiricus, is missing. I suggest in the comments how this can be addressed, basically to add it to the dataset or to use sequences published by other authors for comparison, although on a part of the genome (see Kononov et al. not cited in this ms).

2. the objectives and the discussion should be more clear about the taxonomic level to consider. I would suggest to reduce/remove the part on the family and superfamily level and to concentrate on the Dendrolimus genus. This will help to make the paper more focused, especially if D. sibiricus is added.

Experimental design

The experimental design is fine, although one important species is missing and should be added (suggestions in the pdf file)

Validity of the findings

Results should be integrated with the addition of the species mentioned above.

Additional comments

I think this ms can be considered after the recommendations are addressed.

Reviewer 3 ·

Basic reporting

It is suggested to refer to more papers for the results of phylogenetic tree construction, and discuss the differences between species and the reasons.

Experimental design

no comment

Validity of the findings

In the results and discussion, the similarities and differences between the six species of pine caterpillar based on the base composition, codon, tRNA structure and other aspects and the results of phylogenetic tree are needed more summarize. Codon usage and RSCU were similar among the different Dendrolimus species, and the Figure 3 and 4 seems uninformative. I suggest move these two figures to the supplement material or change them as comparing the Codon usage and RSCU between Lasiocampoidea and Bombycoidea.

Additional comments

Qin et. al. sequenced mitogenomes of three Dendrolimus species, and performed a very detailed analysis of the mitogenomes known in Dendrolimus spp. Based on mitochondrial genome sequences, the paper compared the base compasition, the order of the 37 genes and protein gene of initiation codon and termination codon, the characteristics of A+T rich area structure, and phylogenetic tree was constructed. The information of mitochondrial genome in lepidoptera was enriched, and the authors illustrated interspecific relationship of the six species.
1. In the Background part of Abstract, The authors should add one sentences to explain the discrimination challenging of different Dendrolimus species.
2. Line 66: The reference were cited as numbers here but name and year information in other sentences.

---

## Round 0.2 · accepted · Accept

Dear Dr. Qin and colleagues:

Thanks for revising your manuscript to PeerJ, and for addressing the concerns raised by the reviewers. I now believe that your manuscript is suitable for publication. Congratulations! I look forward to seeing this work in print, and I anticipate it being an important resource for research on pine moth mitogenomics and systematics. Thanks again for choosing PeerJ to publish such important work.

-joe